# A New Approach for Understanding International Hospital Bed Numbers and Application to Local Area Bed Demand and Capacity Planning

**DOI:** 10.3390/ijerph21081035

**Published:** 2024-08-06

**Authors:** Rodney P. Jones

**Affiliations:** Healthcare Analysis & Forecasting, Wantage OX12 0NE, Oxfordshire, UK; hcaf_rod@yahoo.co.uk

**Keywords:** hospital beds, bed modeling, bed occupancy, age-standardized mortality rate (ASMR), crude mortality rate, population deprivation, births, pediatric care, adult care, social care, elderly care, critical care, sex, age, Northern Territory, Tasmania, England, USA

## Abstract

Three models/methods are given to understand the extreme international variation in available and occupied hospital bed numbers. These models/methods all rely on readily available data. In the first, occupied beds (rather than available beds) are used to measure the expressed demand for hospital beds. The expressed occupied bed demand for three countries was in the order Australia > England > USA. Next, the age-standardized mortality rate (ASMR) has dual functions. Less developed countries/regions have low access to healthcare, which results in high ASMR, or a negative slope between ASMR versus available/occupied beds. In the more developed countries, high ASMR can also be used to measure the ‘need’ for healthcare (including occupied beds), a positive slope among various social (wealth/lifestyle) groups, which will include Indigenous peoples. In England, a 100-unit increase in ASMR (European Standard population) leads to a 15.3–30.7 (feasible range) unit increase in occupied beds per 1000 deaths. Higher ASMR shows why the Australian states of the Northern Territory and Tasmania have an intrinsic higher bed demand. The USA has a high relative ASMR (for a developed/wealthy country) because healthcare is not universal in the widest sense. Lastly, a method for benchmarking the whole hospital’s average bed occupancy which enables them to run at optimum efficiency and safety. English hospitals operate at highly disruptive and unsafe levels of bed occupancy, manifesting as high ‘turn-away’. Turn-away implies bed unavailability for the next arriving patient. In the case of occupied beds, the slope of the relationship between occupied beds per 1000 deaths and deaths per 1000 population shows a power law function. Scatter around the trend line arising from year-to-year fluctuations in occupied beds per 1000 deaths, ASMR, deaths per 1000 population, changes in the number of persons hidden in the elective, outpatient and diagnostic waiting lists, and local area variation in births affecting maternity, neonatal, and pediatric bed demand. Additional variation will arise from differences in the level of local funding for social care, especially elderly care. The problems associated with crafting effective bed planning are illustrated using the English NHS as an example.

## 1. Introduction

Defining the optimum level of hospital beds has been controversial [1]. Bed supply can be influenced by multiple factors unique to each country [2,3]. Over the past 30 years, I have been involved in the long-term forecasting of healthcare demand and capacity planning. This has resulted in over 300 publications on these topics and the related issue of financial risk in healthcare purchasing. To avoid self-citation, these papers can be found in the Supplementary file (https://www.mdpi.com/1660-4601/20/24/7171/s1), (accessed on 30 July 2024), which is a recent review of internationally available bed supply [1]. These publications are in alphabetical sections with numbered papers, hence, L.1. is in the hospital bed planning and occupancy section dealing with interpreting daily emergency admissions and discharges, and their impact on occupied beds. These will occur in the text as see L.1, K.7, etc., in [1].

The review concluded that a new model could be applied which plots the ratio of available beds per 1000 deaths against the natural log of the number of deaths per 1000 population [1]. The ratio of deaths per 1000 population is used to approximate the population age structure. It was also proposed that the age-standardized mortality rate (ASMR) represented a widely available measure of population ‘deprivation’ [1]. ASMR could then be used to adjust for the higher/lower bed demand arising from relative population deprivation between countries and local areas.

While benchmarking bed supply using a standard methodology is useful, it does not provide a fundamental measure of the population-level expressed bed demand as occupied beds does [2]. From a planning perspective, it is the ability to benchmark the expressed occupied bed demand, which is the foundation of the ultimate calculation of bed supply.

Regarding the issue of average bed occupancy and bed supply, it has been commonly stated that 85% average occupancy represents the ‘optimum’ hospital average occupancy. However, no one can locate the original study—and 85% is merely a blindly repeated myth [4]. Tools such as queuing theory give far greater insight [5,6,7,8,9]. There are thousands of published studies demonstrating how queuing theory and related simulation studies can be directly applied to understanding hospital bed numbers and occupancy, for example, see [10,11,12,13].

Some years ago, I developed a method to use the Erlang B equation, the simplest form of queuing theory, to show lines of constant turn-away on a graph with available beds as the x-axis and average occupancy on the y-axis; see L.2–L.6, L.12, in [1]. Turn-away can be interpreted as the chaos, inefficiency, and patient harm engulfing a hospital when there are insufficient beds to service the arriving demand, i.e., canceled elective operations, ambulances queuing outside the emergency department (ED), patients stuck in the ED due to the unavailability of inpatient beds, patients lying on trollies in corridors, inpatients boarding in the incorrect specialty beds, and premature discharge from intensive and inpatient care, all creating high levels of busyness additionally leading to staff stress, dissatisfaction, and poor care; see L.2, L.23, L.31, in [1].

The Erlang B equation only applies to individual bed pools within the hospital, hence, to gynecology, pediatrics, orthopedics, surgery, general medicine, etc. The whole hospital’s average bed occupancy then depends on the number and size of the bed pools within each hospital; see L.12, in [1]. From Erlang B, it is immediately apparent that size affects turn-away which affects the bed supply. Concepts such as ‘sweating’ the assets are likewise counterproductive and even harmful.

My research indicated that at the specialty level in the English NHS, the 3% turn-away line represented the best compromise between capital costs and admission from a waiting list as a buffer between fluctuations in demand and available capacity; see L.3–5, in [1]. This figure was derived by comparing the average occupancies of different-sized specialty bed pools against the lines of turn-away. These lines of constant turn-away were also applied at the whole-hospital level to demonstrate that the hospital average occupancy was lower as size was reduced and that in the USA, a figure of around 75% average occupancy applied to hospitals with around 1000 total beds; see L.12 in [1]. This compares well with the average occupancies reported for OECD countries, where the OECD average before COVID-19 was around 77%, with an interquartile range of 67% to 79% [14].

To address these issues, this study first takes the population-level expressed bed demand, as occupied beds (acute plus mental health and maternity, for both elective and emergency admissions), for the populations of English National Health Service (NHS) Clinical Commissioning Groups (CCGs) and matches this with the ASMR for each CCG to derive the value of the slope between ASMR and bed demand. Each CCG is approximately contiguous with one or more local authorities. Using their NHS number, all residents of England are linked to the list of a group of primary care physicians called a GP practice, associated with a CCG. Issues regarding the scatter around the trend line are discussed from the viewpoint of the structural factors that influence bed demand.

Translation of this expressed bed demand into appropriate (efficient, safe) bed supply is also examined using the Erlang B Equation (queuing theory) applied to English NHS Hospital Trusts. Each NHS hospital Trust, some of which specialize in one specific area, i.e., pediatrics, oncology, neurology, orthopedics, ophthalmology, cardiothoracic, etc., can operate from one or more sites and receive patients from multiple CCGs, which comprises the expressed bed demand. The average bed occupancy is then used to determine which Trusts have sufficient beds for the uninterrupted admission of patients into the correct bed pool appropriate to their age, sex, condition, and specialty treatment.

These two components form the basis for the international benchmarking of bed demand and supply and explain why average bed occupancy varies greatly between countries [14]. Hence, a standard set of tools to make international comparisons and to guide regional and local capacity planning is needed.

## 2. Materials and Methods

### 2.1. Occupied Beds in English CCGs

Occupied bed days in the 2014/15 to 2019/20 financial years (ending on 31 March) are from Hospital Episode Statistics (admitted patient care) [15]. Occupied bed days are converted to occupied beds by division by 365 (days per year). To account for same-day admissions, an average length of stay of eight hours is assumed.

### 2.2. English CCG Populations and Age Standardized Mortality Rate (ASMR)

CCG populations for 2019 are from the Office for National Statistics (ONS) [16]. ASMR for CCGs is not widely available. However, most CCGs are approximately coincident with one or more local authorities for whom deaths, crude mortality rate, and ASMR are available [17]. When a CCG comprises two or more local authority areas, the crude mortality rate and the ASMR are calculated as the population-weighted average. Additional ASMR values were obtained by correlation from avoidable mortality for English CCGs in 2019 from the ONS [18]. Avoidable CCG mortality (ASMR) was correlated against all-cause ASMR data for matching local authorities [18]; see Appendix A. In Appendix A, CCG avoidable ASMR was first extrapolated to 2019 by linear regression of the avoidable ASMR between 2011 and 2019. This was then matched against the local authority ASMR in 2019. Some 46 CCGs had a matching local authority ASMR. The correlation in Appendix A was then used to estimate CCG all-cause ASMR where no matching local authority was available.

### 2.3. Available and Occupied Beds in English NHS Hospitals

Hospital available and occupied beds (at 8 a.m.) during the winter of 2023/24 are from daily SITREPS collected by NHS England (20 November 2023 to 17 March 2024) [19]. The data covers all 135 NHS acute Trusts and includes total beds plus a split between adult and pediatric general beds and adult, child, and neonatal critical care. A sample of data was taken as follows: all seven days for the week commencing 20 November 2023 (November is not considered to be a high emergency care month), two random days chosen from each of the five months (November to March), and additional days in each month where the national average bed occupancy is at a maximum. The sample excluded the period before and after Christmas including the New Year (23 December 2023 to 4 January 2024). In total, the sample involved 22 data points. For each hospital, the following were calculated: the average number of available beds, the average number of occupied beds, and, hence, the average percent bed occupancy and the standard deviation associated with the percent occupancy.

Additional bed occupancy data (midnight) was obtained for maternity and mental illness from the NHS England website for the average quarterly occupancy at NHS organizations [20]. The average from the first and third quarters of the 2023/24 financial year was used for available beds and occupancy.

A Freedom of Information request was also sent to the 35 largest NHS Trusts to obtain occupancy data at individual sites for pediatrics, maternity, and all-specialty adult inpatients from November 2023 to March 2024 (as for the SITREPS data series). NHS Trusts can operate from 1 to 6 sites.

Lines of constant turn-away from the Erlang B equation (queuing theory) are from the author’s calculations; see L.12, in [1]. Appendix A provides available beds, average occupancy, and the lines of constant turn-away used in this study.

### 2.4. Year-to-Year Volatility in Occupied Beds for English CCGs

Occupied beds for NHS hospitals are from Hospital Episode Statistics (admitted patient care) [15]. The forecast expressed bed demand in 2019/20, i.e., the last year before the COVID-19 pandemic, is from a linear correlation over the 6 years from 2014/15 to 2019/20.

The data has been corrected for shifts in bed demand, which are mostly downward and can occur if schemes to reduce admissions are implemented by the CCG. Alternatively, CCGs can limit elective admissions to remain within budget. Such an adjustment involves estimating the size of the shift (as the ratio of occupied beds in the year before divided by occupied beds in the year after) and then applying this ratio to the years before the shift.

Financial year data which is low has also been removed. These reflect data loss due to a computer system failure at a hospital where patients from the CCG are serviced. Each CCG will have one main hospital plus other secondary hospitals.

After adjustments, the slope of the trend in occupied beds was calculated by linear regression with the financial year 2019/20 as the intercept. The standard deviation was calculated after adjusting all years to the 2019/20 equivalent, i.e., actual value plus n years times the slope of the trend. The calculated standard deviation is expressed as a percentage of the 2019/20 forecast value.

### 2.5. The Slope of the Relationship between ASMR and Occupied Bed Demand

The power law function describing the relationship between occupied beds per 1000 deaths and deaths per 1000 population demonstrated in this study was used to calculate the magnitude of the distance between the actual beds per 1000 deaths for each CCG and that predicted by the trend line. The difference between actual and predicted values was then plotted against the ASMR for each CCG and a linear trend line was applied. Three such trend lines were calculated, namely using the raw data (which is not equally spaced), using the median value of the data assembled into bins at 100 units of ASMR, and by visually estimating the slope of the line which corresponds to the upper and lower limits of the data. The latter is not a standard statistical method but is used to give an estimate of the likely maximum upper limit to the slope.

### 2.6. Calculation of Excess Winter Mortality

Excess winter mortality (EWM) is calculated based on the average deaths in the 4 ‘winter’ months versus the 8 ‘non-winter’ months. It is a good indicator of acute hospital admissions and bed demand during the winter; see H.1–H.12 in [1]. Data for monthly deaths in England and Wales from 1949 onward was provided by request from the Office for National Statistics (ONS) supplemented by monthly data since 2010, also from the ONS [21], which also covers deaths at the local authority level. The data covering England and Wales was used to calculate EWM from the winter of 1949/50 onwards in Figure A1 in Appendix B, and the local authority level data was used to calculate the EWM for the winter of 2023/24 in Figure A2 in Appendix B.

### 2.7. Moving 12-Month Total of Births in England and Wales

Monthly data for live births in England and Wales from 1980 onward is from the ONS [22]. Monthly data was converted into a moving 12-month total as shown in Figure A3 in Appendix B.

### 2.8. Proportion of Beds Occupied by Same-Day Stay Admissions in 2020/21

Data is from Hospital Episode Statistics at the specialty level [15] and excludes elective day case admissions. Each same-day stay is assumed to have an 8 h stay. Most of these same-day admissions will be emergency admissions to various short-stay assessment units. These data were used to construct Table A1 in Appendix B.

## 3. Results

The results section is split into two halves. Firstly, I aim to investigate the primary expressed bed demand in English CCGs, including the intrinsic volatility associated with the expressed bed demand. Then, I aim to investigate how CCG population ASMR may affect this expressed bed demand, and finally to compare these using wider data from Australia and the USA.

The next section investigates how hospital average bed occupancy can be used as a diagnostic tool to reveal when bed supply is insufficient to deal with the expressed bed demand. Data from English NHS hospitals is used to illustrate examples of sufficient and deficient bed supply. The implications for a whole hospital bed occupancy figure are explored, with the often-quoted 85% occupancy figure being shown to be entirely inappropriate to the field of bed planning.

### 3.1. Factors Regulating the Expressed Bed Demand

#### The Volatility Associated with Expressed Bed Demand

Figure 1 shows the volatility (standard deviation for the 6 years 2014/15 to 2019/20) as a percent of bed demand in 2019/20 for each CCG.

In Figure 1, there is a power law trend to lower volatility as size increases; however, the relationship has a low value for R-squared, indicating multiple sources of variation. Note that in some CCGs, expressed bed demand will be suppressed due to the accumulation of patients onto growing outpatient and inpatient waiting lists [23].

### 3.2. The Relationship Describing Expressed Bed Demand

Figure 2 shows the relationship between occupied beds per 1000 deaths and deaths per 1000 population for English CCG populations as per the approach used for the study on international bed supply [1]. In the previous study, the data for international bed supply was assumed to follow a logarithmic relationship [1], hence the two log relationship lines in Figure 2.

However, in Figure 2, a power law function better describes the actual relationship for occupied beds. Note how the previously used log relationships work well on either side of 8 deaths per 1000 population. The issue of data availability for international beds will be covered in the discussion. Hence, the power law relationship was used to calculate the difference between the actual and the trend line to investigate the role of ASMR in Figure 3.

### 3.3. The Relationship between ASMR and Expressed Bed Demand

The previous study suggested that ASMR was an internationally comparable way to compare intrinsic bed demand in countries with different levels of population ‘deprivation’ [1]. To estimate the additional bed demand in more ‘deprived’ populations, Figure 3 shows the relationship between the difference for the actual minus the expected trend line for total CCG bed demand (acute + mental health + maternity) as beds per 1000 deaths and the all-cause ASMR for each CCG. Three trend lines are shown, all with an R-squared of 0.2528 but with different fixed intercepts. The black dotted line gives the actual trend, the red dotted line shows the relationship when the median value of the raw data is placed into bins at intervals of 100 units of ASMR, and the green dotted line is constrained to the slope describing the upper and lower edges of the data.

The slope lies between 15.3 to 20.3 occupied beds per 1000 deaths per 100 unit increase in ASMR, up to a maximum possible of 30.7. This value is compared with previous estimates in the Discussion as are the reasons for the high scatter around the trend line.

### 3.4. ASMR and Differences in Bed Demand between Australian States

Previous studies, see L.46 and L.57 in [1], demonstrated that the Australian States of Tasmania and the Northern Territory have far fewer beds than the other States despite higher expected bed demand. To this end, Figure 4 shows that both States have appreciably higher ASMRs than the others.

The average ASMR for Australia is in light blue. Hence, ASMR provides an objective basis to explain why more beds are needed in these States, especially in the Northern Territory. Each State appears to have its method for calculating bed requirements, with no way of knowing how well these methods work in the real world. Volatility in bed demand was demonstrated in Figure 1 for English CCGs. Figure 5 investigates the volatility in the ASMR for Tasmania and the Northern Territory.

### 3.5. Applying the Power Law Model to Bed Demand in Other Countries

Having suggested that a power law function applies to expressed bed demand in England, the output of two previous studies relating to expressed acute bed demand, see L.51, L.57 in [1], has been reanalyzed in Figure 6.

There are unavoidable differences in the definition of acute care. The data from England contains occupied beds for maternity and mental illness, elevating bed demand by around 22% over acute demand [15]. The data from Australia and the USA includes all hospital types, including privately owned. This is omitted from the data for England, where private hospitals only cover elective surgical admissions and around 10% more occupied beds can be assumed. Hence, move the line downward by around 12% for England. English and Australian data include an allowance for bed occupancy due to same-day admissions assuming an 8 h average stay. This is omitted from the US data which can be increased by 15%, assuming the USA has more same-day activity than England [15]. This brings the line for the USA closer to England.

The range of deaths per 1000 population on the x-axis reflects the populations covered by English CCGs versus the larger Australian and US states. Note the relatively younger population in Australia.

During the reanalysis of the Australian data, it was noted that the point for the Northern Territory had an undue effect on the power law function. Hence, Figure 6 shows the resulting power law function with the Northern Territory included and excluded.

The Northern Territory is not only an ASMR outlier but also contains most of the Australian population of Indigenous people, who have unique profiles for disease and expressed health care demand; see L.57, in [1]. Hence, the power law line excluding the Northern Territory looks more likely.

### 3.6. Benchmarking Whole Hospital Average Occupancy

This section explores bed occupancy in several contexts such as specialist hospitals, single specialties (maternity, pediatrics), and critical care. The conclusions of these specific cases are then applied to the whole hospital’s average occupancy and an explanation of why average bed occupancy figures for different countries show high variation.

#### 3.6.1. English Specialist Hospitals

Before investigating the issue of the whole hospital’s average occupancy, it is useful to observe the average bed occupancy in specialist hospitals which approximate a single specialty bed pool, i.e., all beds are approximately accessible to any patient.

Figure 7 investigates the reality of specialist hospital occupancy in the light of queuing theory.

This data comes from English NHS Hospital Trusts and is from random 8 a.m. occupancies between November 2023 and March 2024. The lines of ‘turn-away’ are from the Erlang B equation, which assumes that if a server (bed) is unavailable, the person goes elsewhere. The server (bed) must provide an identical service to all other servers (beds).

Erlang B contains several assumptions; however, hospitals requiring unhindered access, such as children’s, oncology, etc., generally need to operate near or below the 0.1% turn-away line. Data for 17 single-site specialist hospitals are shown in Figure 7 and all but one lie below the 3% turn-away line and mostly near or below the 0.1% turn-away line, with 3 below the 0.01% turn-away line, implying instant access under all possible demand scenarios.

The beds in these specialist hospitals approximately meet the criteria of an identical ‘server’, and hence Erlang B is entirely applicable. These hospitals include specialist eye, pediatric, women’s, cancer, orthopedic, and rehabilitation hospitals. Also shown in this figure is the average occupancy plus the standard deviation associated with the daily values.

Figure 7 indicates that the so-called 85% occupancy rule has no application to the real world. Based on the data for the 17 specialist hospitals, it is possible to simulate the whole hospital occupancy. Several scenarios were constructed and are shown in Table 1.

As seen in the simulations, the average for whole hospital bed occupancy in the medium to large hospital scenarios ranges from 77.5% to 82.4%. The first two scenarios are somewhat unrealistic because the largest single-site acute hospital in England has 1200 acute beds, and a large hospital with just four specialties is unlikely. The small hospital scenario has low average occupancy. This closely approximates the results of a study of bed occupancies in US hospitals, see A.7. in [1], and lies within the range of occupancies reported for OECD countries [14]. This gives us a robust benchmark for the international comparison of bed occupancy. The small hospital scenario shows why one-quarter of OECD countries have an average occupancy below 67%—such countries will have a far higher proportion of smaller hospitals, probably due to lower population density.

We can now apply this robust benchmark to English NHS hospitals, known to have exceptionally high bed occupancies from poor policy implementation based on the entirely unsupported political view that England has far too many hospital beds [1].

#### 3.6.2. English Pediatric and Maternity Departments

Pediatric and maternity bed occupancy for the separate hospital sites of NHS Trusts is presented in Figure 8. Pediatric and maternity data for the 35 largest NHS Trusts was provided for the individual hospital sites within each Trust. The website for other Trusts containing the word ‘hospitals’ was also searched to see if maternity was provided at multiple sites. The number of available beds was divided by the number of sites to reflect the smaller actual size, thus moving the data point to the left.

Over half of maternity units operate below the 0.01% turn-away line, indicating immediate access under all demand scenarios. The situation for children’s hospitals/departments shows that over half operate above 1% turn-away.

NHS Trusts are largely free to allocate available beds as they choose, and consequently, both specialties show a wide range of bed occupancies across England. Since both specialties require rapid access, bed occupancies above the 0.1% turn-away line represent evidence of poor bed supply, which is alarming.

#### 3.6.3. English Critical Care Units

The applicability of Erlang B to critical care was established in the 1990s [4]. Figure 9 shows bed number and occupancy data for English NHS Trusts during the winter of 2023/24. This data comes from the daily winter SITREPS [19] and is therefore collected at 8 a.m. rather than midnight. The data is at the whole NHS Trust level but is supplemented with a few data points supplied for individual sites. Children’s critical care tends to only occur at regional centers. Adult and neonatal critical care for the larger NHS Trusts will be split across two or three sites. Hence, the effective bed pool size on these occasions can be divided by 2 or 3 (moved to the left), leading to higher real-world turn-away.

#### 3.6.4. English Adult Acute and Mental Illness Departments

In Figure 10, the data for the 35 largest NHS Trusts is disaggregated to the individual sites run by the Trust and for adults, i.e., excluding pediatrics. The data for Mental Illness is at the Trust level and therefore covers multiple sites and exclusive bed pools, i.e., secure units, children’s, geriatric, male/female, etc.

The whole-hospital occupancy is the weighted average of the occupancies applicable to the constituent specialty bed pools which will be appreciably smaller, implying far higher turn-away than seems to be the case at the whole-hospital level.

For example, the lowest average occupancy data point in the group of the 10 largest NHS Trusts is for the Northern Care Trust (average occupancy of 88%), which roughly services the northern parts of Greater Manchester. It operates from four sites with approximately 380 beds per site. Each site will have further sub-specialization. The second largest Trust is the Manchester University Trust (average occupancy of 92%) which operates from 10 sites including specialist hospitals for Dentistry, Children, and Women including childbirth. The lower occupancy applicable to these smaller specialist sites reduces the overall average apparent occupancy.

Data for psychiatric hospitals is at the NHS Trust level, and hence is for multiple sites. All data should therefore be moved to the left to approximate the real-world situation. Turn-away in psychiatric hospitals is alarmingly high and has been so for many years; see L.2, L.23 in [1].

## 4. Discussion

### 4.1. Are Hospital Beds an Indispensable Asset or a Liability

Before conducting a wider discussion, it is important to discern how policy views beds—are they an indispensable asset to efficiency or a cost to be ruthlessly reduced? A WHO policy brief published in 2004 demonstrated significant reductions in acute hospital bed numbers between 1990 and 2002. It was implied that implementing appropriate policies could lead to further reductions in bed numbers and costs [2]. This policy brief highlighted that the issue of bed demand per se was poorly understood, and no mention was made regarding queuing theory and the effect of hospital size and bed occupancy on turn-away.

It is apposite to illustrate how government policy can hinder the proper understanding of real bed demand. In England, as with all governments, there are Treasury rules regarding the affordability of publicly funded infrastructure [1]. Such rules are entirely appropriate and well-intended. However, these rules were framed from a pure ‘profit and loss’ perspective and do not incorporate any realization regarding queuing theory and turn-away. These affordability rules became highly problematic during the period of the private funding initiative (PFI) which ran from the 1990s to 2018 when it was eventually abandoned as a ‘fiscal risk’ [25].

As stated, the Treasury rules were developed in splendid isolation from the realities of the ‘rules’ governing hospital bed demand and occupancy. They became so divorced from reality that it was impossible to obtain approval for a hospital project without resorting to large-scale manipulation of the figures in the business case [1]. The only way to get a business case approved was to ‘show’ large reductions in bed numbers. This is a self-defeating process, since the calculated future admissions, length of stay, and occupancy bore no relation to reality. External Management Consultants were employed to construct, some may say concoct, these business cases so that they had the surface appearance of validity. Treasury experts would not have the detailed knowledge of the factors behind admissions, length of stay, and bed occupancy to know that the business cases were worthless [1]. However, these business cases were duly approved, and construction of woefully under-resourced hospitals commenced.

Hence, by the winter of 2023/24, the result was the outcomes revealed in this study. It cannot be overemphasized that the winter of 2023/24 was the seventh lowest for excess winter mortality (EWM) in 75 years (Figure A1 and Figure A2 in the Appendix B) [21]. EWM is a direct indicator of adult bed demand via the nearness to death (NTD) effect, see H.1–H.12 in [1], and by the fact that any agent precipitating death will lead to many times more admissions not ending in death [1].

It is alarming that one of the most innocuous winters in 75 years is accompanied by average adult bed occupancies above 95% for almost the entire adult bed complement in England. Every smaller adult bed pool, i.e., trauma and orthopedics, surgery, urology, ENT, gynecology, etc., will be at intolerable occupancy and very high levels of turn-away, organizational chaos, busyness, and unintended patient harm.

Beds should not be viewed as a cost to be avoided but an absolute requirement for an effective and efficient hospital operating at its minimum cost and of patient harm.

### 4.2. The Principle of Variation and Resource Allocation

Understanding the principle of variation is central to understanding resource allocation in healthcare, see N.1–N.39 in [1], especially the optimum hospital bed occupancy issue. Figure 1 is intended to illustrate the inherent complexity behind hospital bed provision. CCGs at the lower boundary in Figure 1 will be serviced by hospitals with severe bed shortages, limiting fluctuation in the number of occupied beds. The CCGs serviced by hospitals with bed shortages will have lower ‘expressed’ occupied bed demand due to patients diverted into a growing waiting list for elective surgery [23].

The high scatter around the trend line reflects intrinsic variable system factors such as population density (and associated higher disease transmission in particular years), metrological factors, variations in local air pollution, etc. See L.51 in [1].

Figure 5 illustrates an additional source of volatility in bed demand Volatility in ASMR implies volatility in incremental bed demand and hence has implications for the average occupancy margin applicable to the hospitals in particular locations, i.e., the margin between bed supply and expressed bed demand. In Figure 5, the ASMR in 2020 in Australia during the COVID-19 pandemic is low for both States. Australia implemented a strict lockdown process which vastly limited international air travel and movement between States. The transmission of multiple pathogens was probably limited during this period, leading to temporary depression in the ASMR. Indeed, the year-to-year volatility in ASMR is driven by poorly understood single-year-of-age patterns; see H.12, Q.6 in [1].

Such high volatility between years is intrinsic to understanding why hospitals must have an adequate average bed occupancy margin to ensure optimum patient flow [1]. CCGs or any other type of health purchaser with the highest volatility in bed demand will experience the greatest difficulty in achieving financial balance. See N.1–N.39 in [1].

The higher volatility in smaller CCGs in Figure 1 is also partly due to Poisson-based statistical variation in admissions which can be approximated as the square root of the number of occupied beds. For context, 200 occupied beds are equivalent to a small hospital, while the largest acute site in England has around 1100 beds. The largest CCG covers all of Manchester. Regarding the scatter, note that high location-specific variation in annual deaths also occurs, see N.20, in [1], i.e., high volatility is not unique to occupied beds.

Given this high intrinsic volatility, it is preferable to use the trend line between 2014/15 and 2019/20 to establish the average expected occupied bed demand in 2019/20 used in Figure 1 and Figure 2. Note that the 2019/20 financial year finishes just before the onset of the COVID-19 outbreak in the UK.

In partial recognition of the effect of size on year-to-year volatility in bed demand and costs, many of the smaller CCGs were merged into 42 larger Integrated Care Boards (ICBs) in 2022 [26]—despite my earlier research clearly showing that the CCGs were far too small, see N.1 to N.39 in [1], and that some of the ICBs are still too small for financial stability.

### 4.3. Limitations of the International ‘Available’ Bed Supply

The international data for ‘available’ beds used in previous studies was limited to 178 countries grouped into seven categories from ‘poorest’ to ‘gross excess’ [1]. The gross excess category only contains 5 countries, while the others contain 22 to 33 countries. Of necessity, allocation into each category is somewhat arbitrary, although to calculate the slope and intercept of the logarithmic relationship, countries are moved between categories, and the values are recalculated. Too little data lies below 8 deaths per 1000 population to calibrate a power law function. The figure of 8 deaths per 1000 population is where the slope of the power law relationship rapidly increases as per Figure 2. For the world countries, of the three categories containing the developed countries, there are only 12, 16, and 17 countries, respectively, most of which lie above 8 deaths per 1000 population.

Countries also occur in each category for multiple bed supply reasons [2,3]. For example, the Maldives, a popular tourist destination, has two main hospitals in Malé, plus three smaller regional hospitals. As a result, the Maldives falls into what was labeled as the ‘gross excess’ category. Barbados, another popular tourist location, has three main hospitals, a geriatric hospital, and a community hospital, and therefore falls into the ‘very high’ category. It is a moot point to which geriatric and community beds qualify as genuine curative beds. Another example is Monaco, which has several public and private hospitals including a cardiothoracic center used by wealthy non-residents from the wider Mediterranean and Africa. Crude bed supply thus reflects multiple causes including the needs of the resident and the non-resident populations.

When applying a power law relationship, as for English CCGs, it is important to have sufficient data below 8 deaths per 1000 population, simply because any regression is drawn toward the region where the data is most abundant. The English CCG data contains 148 CCGs, of which around 65 CCGs are below this figure.

Hence, to benchmark international ‘available’ beds, the previous logarithmic equation remains sufficiently valid to benchmark against broadly similar bed supply but is probably best applied above 8 beds per 1000 deaths. Clearly, ‘available’ beds do not measure fundamental bed demand, which can only be estimated from data specific to ‘occupied’ beds, hence the rationale for this study.

The 148 data points for ‘occupied’ beds in English CCGs are unique because data originates from the English NHS where every location is subject to the NHS funding formula whose total funding for revenue is limited by parliament. Likewise, the availability of beds is determined by a common set of capital constraints. Relative to the international data on available beds, the scatter around the common trend is greatly constrained.

### 4.4. The Relationship with the Age Standardized Mortality Rate (ASMR)

ASMR is a widely available measure of relative population ‘deprivation’ using the standard world population which ranges from around 325 (Japan/Singapore) to >1000 (most African countries) and up to 2000 in Lesotho and the Central African Republic. The average for OECD countries is around 465 [1]. It is also widely available in all developed countries at the small area level. While country-specific alternate measures of ‘deprivation’ may be available [1], it is preferable to use ASMR so that bed models can be internationally compared. When used to compare bed requirements within a country, a country-specific age profile can be applied for the age standardization process, as shown in Figure 4 and Figure 5 for Australia.

The value for the slope of the relationship between beds per 1000 deaths and ASMR in Figure 3 (15.3 to 30.7 per 100 units of ASMR) lies within the range of 13 to 34, as previously estimated [1]. This range looks to apply to both total and acute beds; however, the relationship for pediatric and maternity beds needs to be determined separately, as should that for individual specialties and between elective and emergency [1].

Figure 4 demonstrate that higher bed demand in the Australian states of Tasmania and the Northern Territory, see L.46, L.57 in [1], is associated with higher ASMR. Figure 5 shows that ASMR has high year-to-year volatility and will partly explain the high volatility in Figure 1. The implication is that bed demand models must incorporate total bed provision adequate for ‘bad’ years. This issue will be covered later.

### 4.5. Differences between Occupied Beds in Australia, England, and the USA

A fascinating study by Brunn et al. [3] investigated why Germany has so many critical care and acute hospital beds. Multiple factors specific to Germany apply and explain why reducing expressed occupied bed demand has been difficult. Specific national factors will apply around the world. 

Figure 6 revealed differences in occupied beds in Australia, England, and the USA. After applying adjustments for counting differences and a common reference point of 8 deaths per 1000 population, we arrive at the following beds per 1000 deaths (adjustments are detailed in Section 3.5):

Australia—336 (reference);

England—267 (303 minus 12% adjustment);

USA—156 (136 plus 15% adjustment).

Despite Australia having far more available beds than England, see L.43, L.46 in [1], the expressed bed demand is only 26% higher. This probably reflects the greater access to elective care among the MEDICARE-insured population but a generally lower bed occupancy in Australia’s higher proportion of smaller hospitals. It should also be noted that the previous study [1] used the trend in occupied beds in England to show that reduced NHS funding in recent years has acted to suppress occupied bed demand, especially in the surgical specialties where demand was simply diverted into a growing waiting list [23]. The expressed bed demand in Germany will likely be significantly higher than Australia [3].

However, the data from the USA reveals 42% lower expressed bed demand compared to England. This does not indicate that the USA has discovered some wonderful way of increasing its population health—indeed the ASMR for the USA is higher than England Australia, the average for World Bank high-income countries, and OECD member countries [1], and only ranks around 44th in the world. The low bed demand is more likely to be the unintended outcome of private for-profit health insurance; namely, even with the advent of Obama Care, the poor still struggle to access acute care due to insurance policy ‘small print’ exclusions and co-payments designed to discourage utilization. This exemplifies the conclusions of the German study [3] that specific national factors drive the expressed occupied bed demand.

Note that the power law function for the USA is relatively flat, i.e., lower bed demand in those states with a higher proportion of children. It is of interest to note that in 2020, deaths of children under 5 years in the USA were 2.5 times higher than in Sweden, and the USA ranks 52nd in the world, which is very poor given its high GDP [27]. Access to primary and secondary care is probably limited compared to other countries. In addition, the hospital average length of stay in US states shows evidence for gross distortions, especially in the poorest states; see K.3 in [1]. Expressed bed demand in the USA is highly likely to be artificially reduced. In conclusion, the method appears to be working and gives consistent answers.

### 4.6. Understanding Hospital Bed Occupancy

#### 4.6.1. Limitations and Advantages of the Erlang Equation

The origins of the supposed 85% average bed occupancy figure appear lost in the mists of time and folklore [4] and have been repeatedly shown to have no relevance to the whole hospital [4,5,6,7,8,9,10,11,12,13]. Queuing theory and related methods are far more relevant.

One such study has been repeatedly misused to argue that the 85% figure is relevant. Bagust et al. [28] conducted a simulation specifically using a 200-bed medical bed pool. Assumptions were applied regarding the weekly profile for admissions, and the summer/winter differential in admissions was set to be artificially low. However, they concluded that for the 200-bed medical pool, immediate access to a bed began to fail between 80% to 85% average occupancy and that bed crises emerged around 90% of average occupancy. We can cross-check these conclusions using Figure 7, where, for 200 available beds, around 0.1% turn-away occurs at 83% average occupancy, 1% turn-away at 90% average occupancy, and 3% turn-away at 93% average occupancy. The two different methods are in good agreement, and 85% average occupancy therefore only applies to a bed pool with 200 beds (with approximately no delay to admission) is are not in any way applicable to the whole hospital’s average bed occupancy.

As with any model, the Erlang B equation (queuing theory) contains several key assumptions, namely the following:The average rate of admissions is the same on a 24/7/365 basis.Erlang assumes a particular distribution for length of stay.Queuing for admission does not occur.

None of these assumptions is universally applicable; hence, why suggest this as a method to compare occupancy? There are several good reasons, namely the following:Erlang B is the simplest of the Erlang equations and calculations are readily available and therefore widely accessible, with several Erlang B calculators freely available on the internet.The output from Erlang B was compared to the real world of hospital bed occupancy in the late 1990s (when hospital beds were more abundant in England) by this author, and 0.1% turn-away was shown to apply to specialties requiring immediate access such as maternity, pediatrics, oncology, and critical care. In the USA, a 0.1% turn-away is also widely applicable since any delay to elective surgery is usually very short. Likewise, 3% turn-away encompassed most other specialties involving a mix of emergency and elective admission from a waiting list, as is usual in the English NHS.

For rapid comparison, there is no need to resort to more complex versions of the Erlang equations. The main point is that the Erlang B equation should be applied at the specialty level and that the whole hospital bed occupancy is the weighted average of all the constituent specialty bed pools.

Bagust et al. [28] highlighted that their simulation only worked for a modest seasonal profile. This study highlighted that the winter of 2023/24 was one of the least ‘bad’ in many years (Figure A1 and Figure A2 in the Appendix B). It is known that in the real world, the volatility associated with hospital admissions is 2 to 3 times higher than from simple Poisson randomness (assumed to apply to Erlang B); see N.1, N.2 in [1]. Does this invalidate the application of Erlang B? Certainly not, it simply implies that the average demand is constantly changing and that Erlang B applies to the ‘current’ average. Hence, what are the number of beds needed to service the expressed demand at various times in the year? See L.1, L.6 in [1].

Finally, are the lines of turn-away precisely accurate? Recall that the data is a five- or six-month average, and during this time, bed demand will fluctuate due to circadian, weekly, and monthly patterns. Also, Trust-supplied data includes Christmas and New Year when occupancy falls to a minimum, thus decreasing the average. However, absolute precision is not required since every hospital can quickly see how they compare to others.

#### 4.6.2. Hospital Busyness and 85% Occupancy

There is one situation where 85% is generally applicable in that from a staff perspective, it is an approximate measure of busyness, see L.23, L.31 in [1], i.e., above 85% occupancy, staff experience unrelenting busyness and therefore are more stressed and more prone to burn-out, neglect of standards for hand washing, and medical errors. Busyness explains why hospital-acquired infections increase above 85% occupancy; see L.23, L.30 in [1]. The effects of busyness will be largely independent of hospital size. 

The literature based on 85% occupancy must be interpreted from the perspective of staff busyness rather than the effects of turn-away upon patients. These are two different processes based on differing staff versus patient perspectives. From Figure 7, Figure 8, Figure 9 and Figure 10, it should be clear that most English hospitals simultaneously experience the deleterious effects of busyness and turn-away, with the patient suffering the consequences of both forces. A literature review on the effects of high bed occupancy conducted by the English National Institute for Clinical Excellence (NICE) in 2018 confirmed the deleterious effects of high occupancy [29]. However, this systematic review omitted any reference to the effect of hospital size on turn-away. The results would therefore need to be re-interpreted by locating the number of beds for the hospitals mentioned in all the published studies and then using Figure 7, Figure 8 and Figure 9 to estimate the turn-away. Turn-away and staffing levels per patient (a better measure of busyness) would then need to be disentangled.

#### 4.6.3. Erlang B and the Real World

As demonstrated in Figure 7, Erlang B widely applies to all specialist NHS hospitals. The only exception was the Clatterbridge Cancer Centre NHS Trust, where it is assumed that sequential outpatient/inpatient treatment (a form of queuing) allows this NHS Trust to operate at the 3% turn-away line.

One of the hospitals with the highest average occupancy, namely the Clatterbridge Cancer Center in Liverpool (88% average occupancy with 101 beds), has recently received additional patients due to a reorganization of cancer services among the Liverpool hospitals. This necessitated higher levels of scheduling for admissions, allowing for operation at slightly higher average occupancy equivalent to the 3% turn-away line. See Hu et al. [30] regarding how an acceptable queuing time can increase throughput. Additional beds are probably justified in coping with the increased bed demand.

Another such hospital is the Royal Papworth in Cambridge, a specialist cardiothoracic and transplant center. Once again, a higher level of scheduled admissions allows for operation at slightly higher average occupancy. The average plus one standard deviation of occupancy at the Royal Papworth is 96.7%, and days with 100% occupancy do occur, implying that several additional beds are probably required to enable the uninterrupted flow of patients. The average plus 1 standard deviation (STDEV) indicates that bed occupancy (midnight) is 1 or more STDEV higher than average on 16% of days and has a higher turn-away.

A point to note from Figure 8, Figure 9 and Figure 10 is that the average occupancy during the winter of 2023/24 in English hospitals is critically high. This is especially so for most pediatric wards in general hospitals. It must be recalled that the winter of 2023/24 was remarkably innocuous for infectious outbreaks [31] and temperature. The situation for pediatric wards can therefore be described as falling into the ‘never event’ category.

Finally, given that hospital demand shows high volatility depending on multiple environmental and infectious stimuli, it is highly advised that hospitals employ software to forecast short-term demand so that staffing costs can be minimized [32,33,34].

##### Maternity and Pediatric Bed Occupancy

Bed occupancy for maternity departments needs to be understood in the wider context of the up/down undulations in births emanating from the World War II baby boom.

During World War II births in England and Wales reached a minimum of 579,000 in 1941. After the cessation of the war, this quickly resulted in a maximum of 881,000 in 1946, a 52% increase. Births subsequently fell to a minimum in 1955 and another maximum in 1974, after that repeating the cycle [22], also C.7 in [1]. Figure A3 in the Appendix B shows the cycle since 1980 and the minimum births during the winter of 2023/24. Maternity bed demand in the winter of 2013/14 should therefore be at its lowest point and bed occupancy should be viewed in this light.

From Figure 8, we see that apart from nine units lying above the 5% turn-away line and two units reportedly lying above the 50% turn-away line, most units lie below the 0.1% turn-away line, and quite a few are below the 0.001% turn-away line. This is consistent with births being at a minimum. Given that the next peak in births is due around 2033, there is no need to close maternity beds since bed occupancy will have increased by around 20%. Indeed, all units currently lying above the 5% turn-away line will need to expand their bed numbers.

The reported percentage bed occupancy for maternity is measured at midnight and this omits patients admitted and discharged into overnight stay beds on the same day. Within the maternity bed pool (Obstetrics + Midwifery), this represents around 12% of bed occupancy. See Table A1 in the Appendix B for a list of specialty same-day occupancy proportions. In Figure 8, this will not represent a problem for those units below the 0.1% turn-away but will become increasingly problematic as the midnight turn-away increases above 5%.

It must be assumed that the management at the units/hospitals above 5% midnight turn-away did not understand the undulating nature of maternity bed demand and may have prematurely closed beds after the last peak in 2012; see C.7 in [1]. I am unaware of any national guidance regarding such trends and their implications for capacity planning (both for staff and bed numbers) in this specialty. It would be interesting to see if maternity units experiencing high turn-away are those associated with reported high levels of harm [35].

As context to Figure 8, it should be noted that the winter of 2023/24 was one of the warmest and wettest winters on record and was somewhat unremarkable for influenza, COVID-19, and Respiratory Syncytial Virus (RSV) [31]—the low levels of respiratory viruses were probably an outcome of the unusual weather conditions. Hence, pediatric bed demand should be far lower than usual, and accordingly, the maximum number of pediatric beds closed due to RSV was only 189 (3% of available beds) on 29 November 2023 [19].

Pediatric bed demand over the 2023/24 winter is therefore of greater importance given the unremarkable levels of viral pathogens normally associated with pediatric respiratory admissions [31]. To emphasize this point, an unremarkable winter is nonetheless characterized by pediatric bed demand incompatible with bed availability.

From Figure 8, given that around 25% of pediatric demand occurs in the first year of life [15], the implication of a minimum in births for 2023/2024 suggests that pediatric bed supply in England is at crisis level. In Pediatrics, some 6% of daytime-occupied beds are from same-day admissions (Table A1 in the Appendix B) [15]. Once again, this will have the greatest impact on daytime bed occupancy in those units above the 5% midnight turn-away line. 

As far as can be discerned, NHS England appears to be unaware of any issues regarding pediatric bed supply—especially given the expected steady rise in births through to 2033.

##### Critical Care Bed Occupancy

England has always had high levels of turn-away in CCUs; see L.2, L.30, L.56 in [1]. While CCU bed numbers are increasing, this seems to be a reactive process. I suspect that turn-away is never discussed lest the public becomes aware of the full extent of the problem. A degree of rationalization of CCU beds in London is possible, see L.30 in [1], to create larger units with lower levels of turn-away. The independence of NHS Trusts largely precludes the necessary cooperation to achieve this goal.

As the term ‘critical care’ implies, the occupancy levels should be around or below the 0.1% turn-away line as is only the case in several hospitals. Turn-away above 20% is unacceptable and would contribute to higher in-hospital mortality [36]. A study published in 2001, see L.2 in [1], gave very similar results to Figure 9. While CCU bed numbers in England have increased, see L.30 in [1], so has demand, leading to no net change over 23 years. CCGs can veto acute hospital plans to expand their CCU bed numbers based purely on cost, possibly ignoring adverse consequences [36].

##### Adult and Psychiatric Beds

Calculation of excess winter mortality (EWM), a proxy for hospital winter bed demand, see H.7 in [1], shows that the winter of 2023/24 had unusually low EWM; the seventh lowest in the past 75 years, i.e., a once-in-a-decade event. See Figure A1 and Figure A2 in the Appendix B.

The levels of common pathogens were unremarkable [31] and the maximum beds occupied by influenza patients was 2481 (2.5% of adult beds) on 5 February 2024. Beds closed due to Norovirus peaked at 971 (1% of available beds) on 11 March 2024 [19]. As suggested by the very low EWM, this was an exceptionally low winter for pathogens. This materially affects the interpretation of Figure 10.

As a context for Figure 10, in 1000-bed hospitals in the USA, the average occupancy is around 78%, implying a sufficient occupancy margin for immediate access to a bed of choice; see L.12 in [1]. The implication in England for whole hospital occupancy lying on the 0.1% turn-away line is that every single bed in the hospital is accessible to any patient irrespective of age, sex, or condition. This would represent operational chaos. Indeed, most acute hospitals in England operate on the verge of chaos, with medical patients ‘outlying’ in other specialty beds, frequently canceled elective operations, and large ‘bed management’ teams devoted to facilitating patient discharge to squeeze in the next patient from the queue for (elective or emergency) admission. This operational chaos is evident in the high proportions of hospitals operating above 50%, 20%, and 5% turn-away. Operating at high occupancy is associated with higher levels of complications in patients ‘stuck’ in the emergency department [37], higher overall and surgical mortality rates and lower health gain [38], truncated length of stay due to bed insufficiency [39], and higher rates of readmission after premature discharge [40,41], while decreasing bed occupancy reduces in-hospital mortality and enables the rapid transition from the emergency department to an inpatient bed [42]. All of these are directly due to turn-away.

##### Long-Stay Patients

While admissions due to common winter pathogens were exceptionally low during the winter of 2023/24, there is a problem with long-stay patients in acute beds. On 30 January 2024, some 51,092 beds (52% of available beds) were occupied by patients with a stay of >7 days. Around 18,000 beds (18% of available beds) are consistently occupied by patients with a stay >21 days [19]. This is likely due to chronic underfunding of adult social care [43]. This problem arises from government policy rather than the NHS per se. Hence, if the government chose to adequately fund adult social care, then NHS bed demand may decline.

The key question is to what extent increased funding would decrease NHS bed occupancy. It has already been noted that nursing home bed capacity is low in England; see L.50 in [1]. The perceived bottleneck may shift to a different place with little real effect.

#### 4.6.4. Implications of Size to Capacity Planning

There are several profound implications of size to capacity planning, with some interesting examples.

A national initiative in Norway required 400+ different-sized municipalities to establish municipal ‘acute’ units (MAUs) to avoid hospital admissions. The goal was to transfer 240,000 patient days to these units, the equivalent of 658 occupied beds—on average just 1.7 occupied beds per municipality. The policy initiative seemingly overlooked the implications of queuing theory and a 24% increase in available MAU beds over the number planned was needed to meet the national target, based on the range in size between municipalities [6]. Figure 7 immediately suggests that this policy initiative would experience problems since an average of 1.7 occupied beds implies 8 available beds at 0.1% turn-away (25.6% average occupancy). This average figure also ignores seasonality in demand.

In Australia, a universal health insurance scheme was implemented in February 1984 which largely covered non-emergency health care [44]. Apart from the increase in transaction costs, the unintended consequence was that elective care was shifted away from large acute sites to a multitude of small private hospitals. Australia now has a high number of available hospital beds; see L.10, L.46 in [1]. The resulting escalation in capital costs was passed onto the insured population via higher insurance premiums. The uninsured population receives elective care with considerable delays, i.e., a large and growing waiting list, from public hospitals.

In the USA, there are two factors. Firstly, the huge geographical size means that a high proportion of the population lives in rural areas [45], see E.6, P.4 in [1], which are serviced by small community hospitals. Due to the high capital costs, these small hospitals are not financially viable and are largely run by the State governments. In addition, healthcare in the USA is largely funded via numerous private for-profit insurance companies, with high transaction costs. Hospitals are run largely by private companies which compete for market share. Hence, in most towns, there will be two or three competitor hospitals when one larger hospital would otherwise be justified. As a result, the USA has a very high proportion of hospitals with fewer than 100 beds; see K.3 in [1]. Once again, the excess capital costs are carried by the insured population.

The population density in the world’s poorest countries is generally low [45], creating additional problems for universal healthcare. Rural and remote parts of developed countries suffer from the same issues.

At a more pragmatic level, the implications of queuing theory and the Erlang equation are that one larger site is preferred over two smaller sites in terms of lower capital costs and the ability to operate at higher average occupancy; see L.3 to L.5 in [1]. When the need for separate male/female wards is factored into the capacity problem, it may make greater sense to have a dedicated female unit for the specialties of Gynecology, Urology, and General Surgery than separate specialty bed pools. On that occasion, the male equivalent will be sized to accommodate just Urology and General Surgery patients. An alternate approach may be to use single rooms to balance the fluctuating needs for single-sex accommodation; see L.15 in [1]. In some locations, maternity and pediatric inpatients may need to be rationalized to one site, with multiple sites covering the outpatient aspects of pregnancy/childbirth care.

Each location must evaluate various options because various types of care show different seasonal patterns, see L.6 in [1], and the intrinsic volatility in demand is location-specific; see N.16 to N.28 in [1].

#### 4.6.5. The Reality of Volatility in Demand versus a Policy-Based View

The final implication is that hospitals should never be sized based on annual averages. Queuing theory clearly states that the point of service should be sized based on the period(s) of maximum demand. This will be especially relevant to those specialties with seasonal peaks such as pediatrics, medicine, and trauma. Hence, hospitals need to be constructed with sufficient beds to cope with surges in demand. These beds may be unstaffed for much of the year but are ‘opened’ or staffed as needed. The bed planning process in England is unfit to meet this purpose, as illustrated in Figure 8, Figure 9 and Figure 10.

It has been my observation for over 30 years that health care policy in England is formulated based on a skewed view of reality. Hence, we (government agencies) ‘know’ how demand ‘should’ or ‘ought’ to behave—failure to match this viewpoint is, therefore, the fault of the NHS due to inefficiency, see L.24 in [1] as an example. Such views are not restricted to politicians, and as observed by Hollnagel [46], ‘*The difference between how work is believed to be carried out, according to plans or the beliefs of people not intimately familiar with the activity, versus how and where it is actually carried out…*’.

The public is constantly informed that the NHS is inefficient, but no one ever says that insufficient beds generate a large part of this inefficiency. Any organization subject to severe resource constraints will exhibit high apparent inefficiency.

As an example, the “Priorities and operational planning guidance 2024/25” released by NHS England [47] emphasizes that process improvement is needed to achieve multiple goals and standards. Nowhere in this document is any reference made to the fact that flow through the emergency department, inpatient beds, and into critical care largely depends on the mismatch between available capacity and presenting demand. This is akin to blaming the NHS for someone else’s problem.

This author has consistently demonstrated that the seasonal and the year-to-year volatility in demand is 2 to 3 times higher than can arise from simple statistical (Poisson-based) randomness; see N.1 to N.39 in [1]. I will now describe some real-world examples.

I have observed that the correct sizing of pediatric units involves constructing daily bed occupancies over the past 10 to 15 years, adjusting these for changes in underlying demand (as per Figure A3 in the Appendix B), and then selecting the daily point of maximum demand. The usual practice for an NHS business case is to take last year’s admissions and extrapolate forward under highly unrealistic assumptions regarding ‘demand management’ via various improvement schemes including length of stay reduction [1]. This only leads to the insufficient bed numbers documented in Figure 8, Figure 9 and Figure 10.

I have also observed that Trauma care follows poorly understood long-term patterns in bed demand. On one occasion, a hospital observed that Trauma bed occupancy was exceptionally high, without any seeming cause. I suggested a Fourier Transform to analyze daily admissions. A Fourier Transform looks for patterns in frequencies such as vibration, sound, etc. The Fourier was duly applied to the daily over 10 years and a set of admission frequencies were observed. While the Fourier ‘understood’ the cause, we were left mystified regarding the explanation. It is now recognized that trauma admissions are the product of complex interactions between seasonal, weather, temporal, age, gender, policy, and social factors [48,49,50,51,52,53].

In addition to these examples, I have published multiple studies demonstrating the spatial movement of unrecognized infectious outbreaks causing large increases in medical admissions; see Q.1 to Q.18, R.1 to R.18 in [1]. The costs associated with such variable demand are likewise highly volatile and location-specific. Associated length of stay likewise shows high volatility and cannot be treated as a value that can be easily manipulated downward; see K.1 to K.9 in [1]. Figure 1 is simply the outworking of such a reality. All have capacity planning implications that are location-specific and mostly ignored when applying annual averages for admissions and length of stay.

This has profound implications to how health care is funded. Many countries have chosen a government-run insurance scheme with the freedom to adjust premiums to reflect this volatility in demand and costs. In England, the NHS is funded via general taxation, which competes with other budgetary pressures. The outworking is that the NHS lurches from feast to famine because the funding does not match the flexibility dictated by volatile demand; see N.1 to N.39 in [1].

As stated in the previous study [1], healthcare policy is profoundly important and can be equally destructive if it is based on policy-based evidence rather than evidence-based policy.

#### 4.6.6. A Turn-Away Based System for Assessing Hospital Efficiency and Effectiveness

It is proposed that the concept of turn-away be widely disseminated to hospitals by government health departments. The equivalent to Figure 7, Figure 8, Figure 9 and Figure 10 can then form the basis for constructing a system for ideal occupancies based on size. Hence, a realistic starting point in England is no greater than 1% turn-away in maternity, pediatric, and CCU units and no greater than 3% turn-away for the totality of the adult bed pool. Lower levels of turn-away are desirable but may not be achievable given current constraints.

#### 4.6.7. The Impact of Flawed Policy on Bed Numbers in England

The ratio of acute occupied beds per 1000 deaths in England has been remarkably constant since 1998/99 [1], when there were 107,729 available overnight beds and 28,697 geriatric beds [20]. Geriatric beds were included in ‘general and acute’ beds in 2010/11. In Q3 of 2023/24, there were 104,455 available general and acute beds. By 2009/10, geriatric beds plateaued at 21,000 available beds, giving a current minimum possible of 87,000 acute beds. Hence, available acute beds have probably declined by up to 20%. In 2010/11, English NHS general and acute bed occupancy was around 86%, steadily rising to 91.6% in Q3 of 2023/24 which is a rise of 7%—the gap is higher because bed occupancy was not reported before 2010/11. Hence, the result of 30 years of flawed bed planning has created a deficit of >7% to <20% acute beds. The difference depends on whether geriatric beds represent curative or palliative beds and how this depends on adult social care funding, which is currently underfunded.

This is reflected in unacceptably high turn-away in nearly all adult acute beds in Figure 10 and many pediatric beds in Figure 8. The gap is now so large that it would take at least a decade to remedy. As Kakad et al. [6] titled their paper, “Erlang could have told you so—A case study of health policy without maths.”.

It is now known that patients parked on a waiting list experience higher fundamental healthcare costs [54], which represents yet another aspect of the deleterious consequences of turn-away.

### 4.7. Future Research

As proposed in the previous study [1], occupied beds must be calculated using real-time length of stay measured in days, hours, and minutes, preferably as days with two decimal places. Occupied beds must also include all same-day stay admissions with real-time LOS. Several research topics regarding expressed bed demand are suggested.
An examination of how to quantify the grey areas regarding the definition of a curative bed. For example, palliative care, aspects of long-term geriatric care, etc. Should patients who die in the hospital be counted separately?Quantification of the proportion of occupied beds with persons staying 0–7 days, 7–14 days, 14–21 days, etc., as an approximate means of determining how the hospital/community interface influences expressed bed demand.Splitting occupied beds into elective and emergency categories.Determining the extent to which certain types of healthcare funding influence the expressed bed demand, i.e., insured vs. uninsured, government-run vs. private health insurance, etc.Greater emphasis on the role of ASMR on expressed bed demand and whether additional factors such as distance to the nearest hospital, ethnicity, social group, etc., are required to qualify further expressed bed demand.A study in Australia regarding expressed bed demand for Indigenous versus non-indigenous people, possibly using Heath Board-level data after adjustment for ASMR.A study in the USA using county-level data after adjustment for ASMR.A wider study among European countries or at the state level in a large country such as Germany.Urgent research required to determine the role of turn-away on the safety of maternity units.

Regarding the lines of constant turn-away to compare bed occupancy, recall that the method is designed for easy and rapid benchmarking rather than getting lost in the detail of more complex methods. The following studies are suggested.
Building a sufficient international reference database will require country-by-country studies. As a minimum, organizations such as the WHO, OECD, etc., should report both available and occupied bed numbers. A suggested simple whole-hospital occupancy data sheet may include total available and occupied beds and the number of major specialty-specific bed pools, i.e., critical care units, pediatrics, trauma and orthopedics, gynecology, etc. Whole-year figures are necessary to avoid the role of seasonality in bed occupancy. However, seasonal profiles are an absolute requirement for local hospital bed planning.Research based on the Erlang equations which incorporate queuing to determine the length of the queue which is beneficial to increased throughput rather than detrimental to performance. For example, see Hu et al. [30].

### 4.8. Limitations of the Study

International comparisons will always involve grey areas around the definition of a ‘curative’ or ‘acute’ bed. Bed demand is volatile and therefore requires data spanning multiple years. The suggested use of excess winter mortality (EWM) to quantify winter bed pressures requires further study. EWM has an approximate implied age profile and does not work well for children and younger adults.

The power law function describing occupied beds is an empirical curve fit, and a more sophisticated formula may be proposed. It has the great benefit of simplicity and hence ease of application.

The calculation of ASMR depends on the weighting assigned to age in the standard population; see Appendix A [55,56]. The WHO World Standard Population (WSP) incorporates a high proportion of children and young adults. The Australian Standard Population is weighted toward middle age, while the European Standard Population is weighted toward older ages. For international comparison, it is suggested that all studies provide ASMR calculated for the WSP, the ESP, and the local standard population.

## 5. Conclusions

Data regarding the expressed bed demand for English CCGs has revealed that the previous formula developed for international ‘available’ beds is better modeled as a power law function. Preliminary evidence shows that a power law function can also be applied to occupied acute beds in Australia and the USA.

Higher ASMR serves a dual role since in less developed countries higher ASMR arises from limited access to healthcare [1]. However, in the local areas within developed countries, it is associated with higher bed demand [1], and ASMR is suggested as the best available measure for both international and intranational comparison. Additional local factors such as urban/rural, weighted or lived population density, nursing home beds, and level of social care funding are likely to be relevant to the expressed bed demand.

Lines of constant turn-away from the Erlang B queuing model are used to determine if individual hospitals have sufficient available beds to meet the expressed bed demand. For optimum patient flow and efficiency, the average occupancy for a 1000-bed hospital should be around 75% to 80%, see L.12, in [1], and lower than this as size reduces. This does not imply that all beds should be staffed but that staffing should be flexed according to patient severity and seasonal variation in demand.

Average hospital occupancy in England is excessively high, with adult, pediatric, and neonatal beds in some hospitals being dangerously high. This is the unfortunate outcome of 30 years of misplaced bed number policy with inappropriate Treasury rules for the affordability of new capital projects [1]. Financial restrictions arising from the 2010 financial crash only exacerbated a poor situation.

Robust tools are now available to compare the interrelated issues of international bed numbers and occupancy. This prevents the manipulation of reality by politicians seeking to make unsupported statements regarding hospital bed demand and supply. It is the totality of the health and social care systems which determine the expressed bed demand. Factors such as population density play an additional role in hospital size and hence the necessary occupancy for optimum patient flow.

## Figures and Tables

**Figure 1 ijerph-21-01035-f001:**
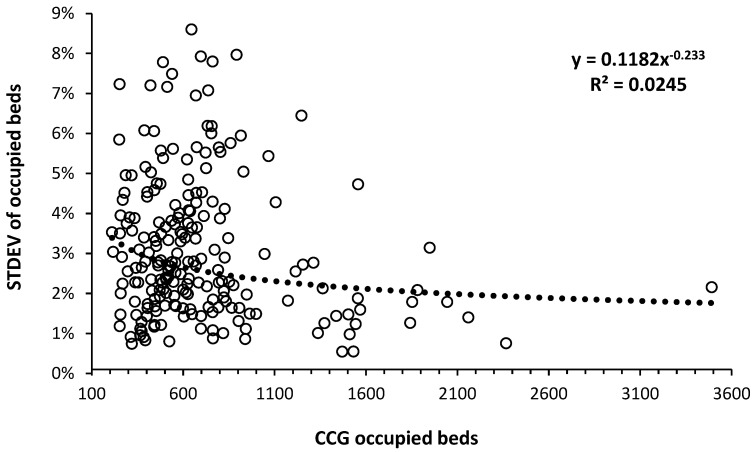
Standard deviation (STDEV) associated with expressed bed demand in English CCGs. STDEV is calculated relative to the 2019/20 average of occupied beds.

**Figure 2 ijerph-21-01035-f002:**
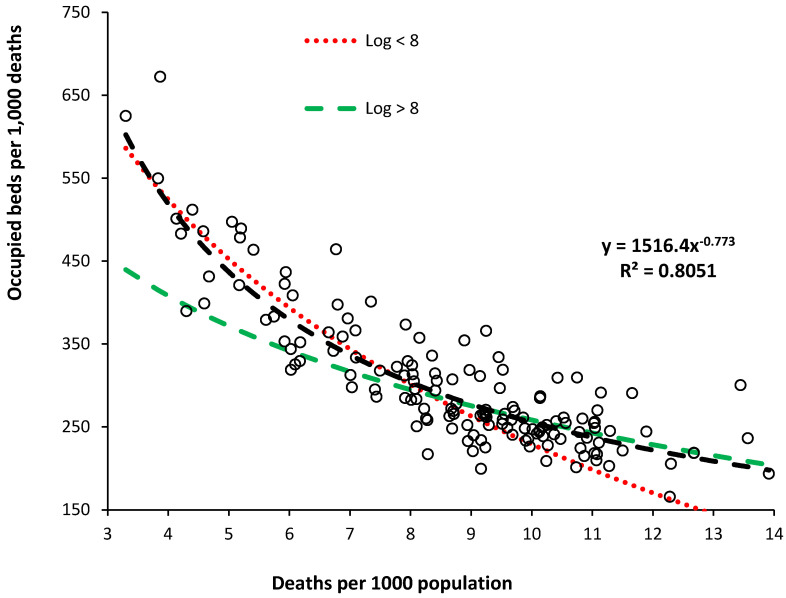
CCG occupied beds (2019/20) per 1000 deaths versus deaths per 1000 population. The data is fitted using two log relationships on either side of 8 deaths per 1000 population and a power law function (black dashed line).

**Figure 3 ijerph-21-01035-f003:**
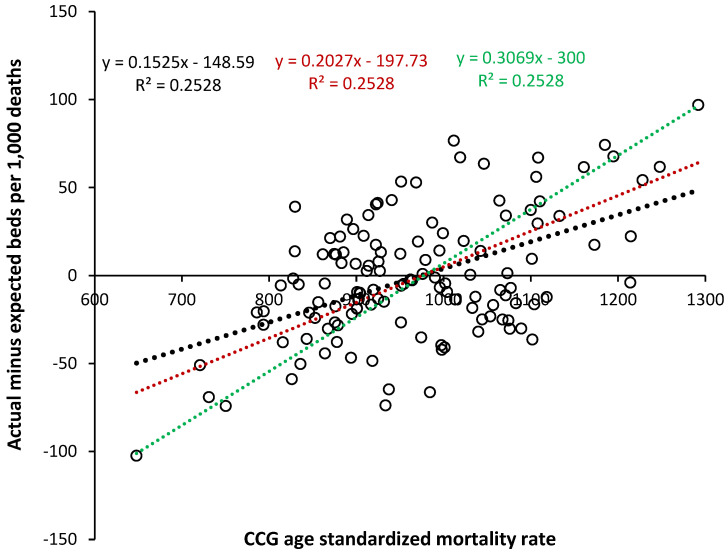
The slope of the relationship between expressed bed demand in English CCGs and the age-standardized mortality rate (ASMR).

**Figure 4 ijerph-21-01035-f004:**
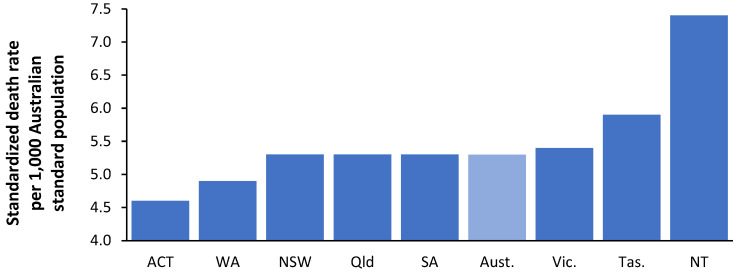
Age-standardized mortality rates for the Australian states in 2019. ASMR in Australia is standardized to the Australian population [24]. Hence, multiply by 73.4 to give the equivalent ASMR using the standard world population. ACT = Australian Capital Territory, WA = Western Australia, NSW = New South Wales, Old = Queensland, Aust = Australia, Vic = Victoria, Tas = Tasmania, NT = Northern Territory.

**Figure 5 ijerph-21-01035-f005:**
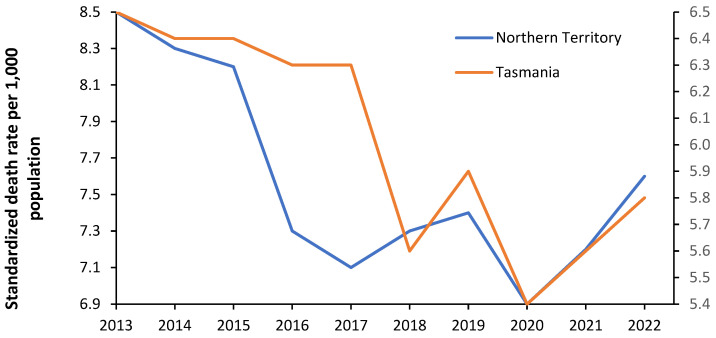
Age-standardized mortality rates for the Australian states of Northern Territory (left axis) and Tasmania (right axis) from 2013 to 2022 [24]. ASMR is to the standard Australian population.

**Figure 6 ijerph-21-01035-f006:**
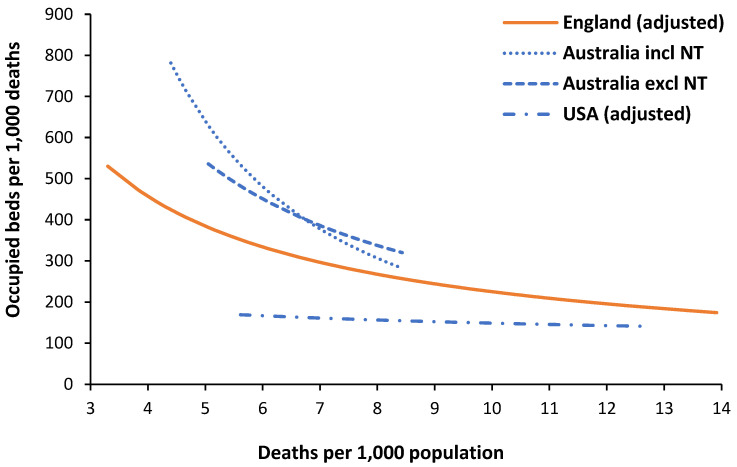
Comparison of occupied beds per 1000 deaths between England (adjusted), the states of Australia, and the USA (adjusted). NT = Northern Territory.

**Figure 7 ijerph-21-01035-f007:**
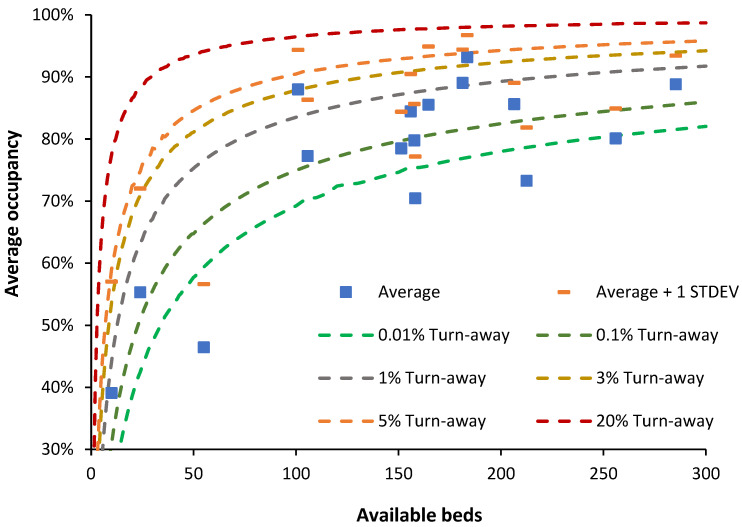
Whole-hospital occupancy for English specialist hospitals (single specialty).

**Figure 8 ijerph-21-01035-f008:**
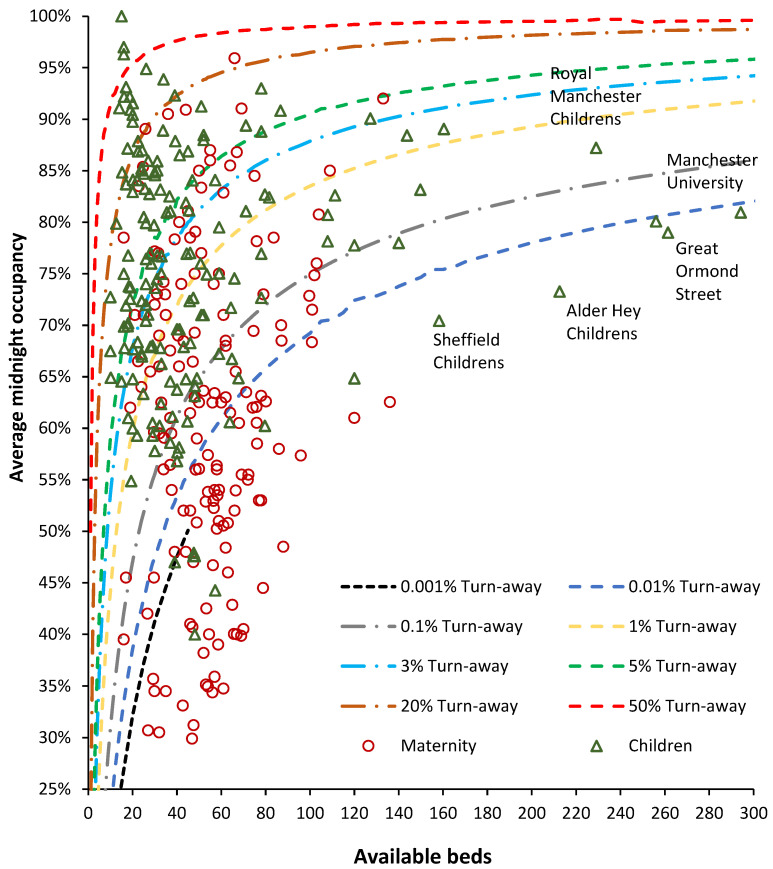
Single-site bed numbers and occupancy for children’s and maternity departments for English NHS Trusts.

**Figure 9 ijerph-21-01035-f009:**
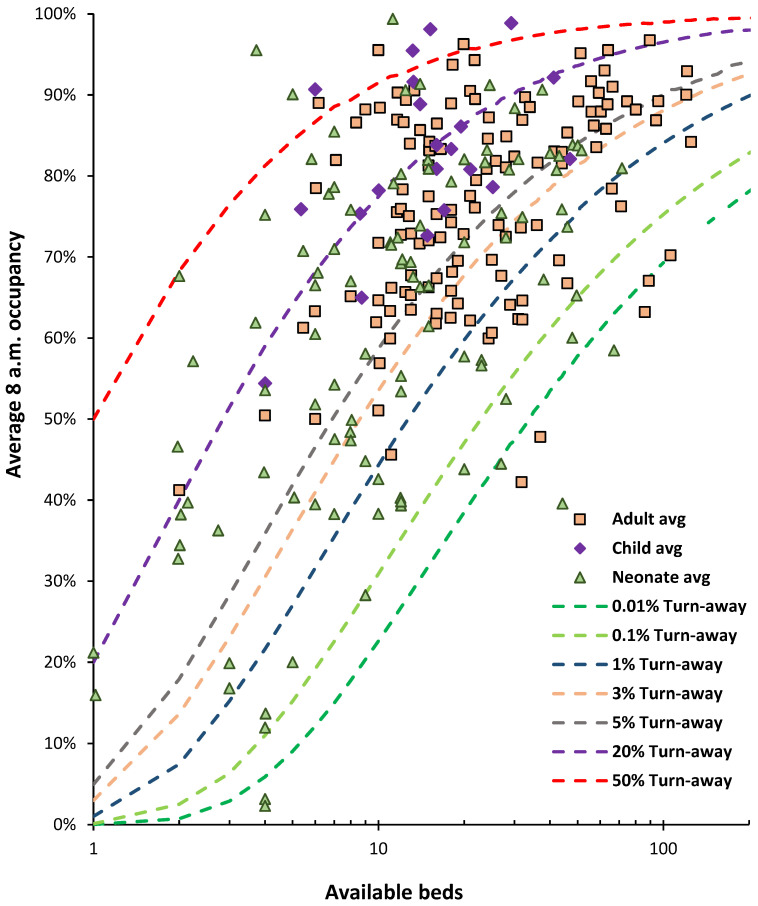
Average (staffed) bed numbers and occupancy for adult, children’s, and neonatal critical care units during the winter of 2023/24 in English NHS Trusts.

**Figure 10 ijerph-21-01035-f010:**
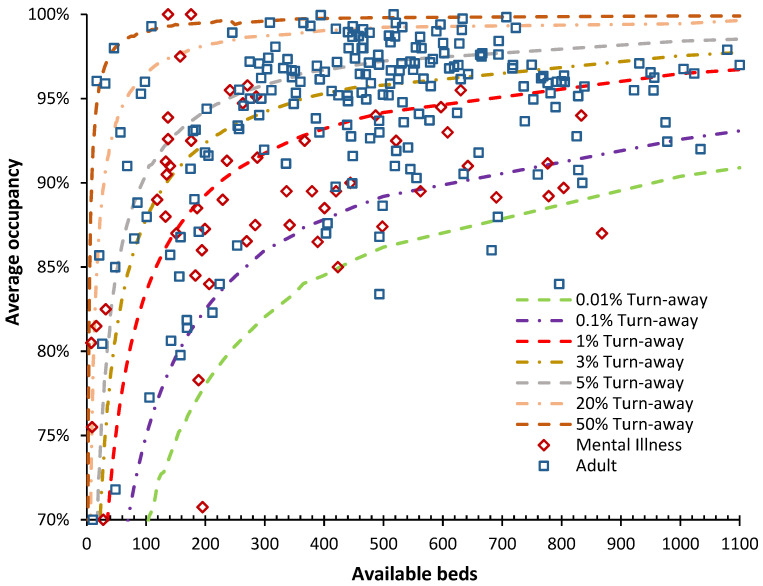
Average (staffed) bed numbers and occupancy for adult acute care (winter of 2023/24), and psychiatric hospitals (Q1 and Q3 in 2023/24).

**Table 1 ijerph-21-01035-t001:** A real-world simulation of total bed number and average occupancy in a properly resourced acute hospital that facilitates unhindered patient flow and hence optimum efficiency.

Scenario	Total Beds	Average Occupancy	Comment
A very large 16-specialty hospital	2409	81.6%	All 16 specialist hospitals combined
A large 4-specialty hospital	960	82.4%	The 4 largest specialist hospitals
A medium to large 9-specialty hospital	761	77.5%	The 9 smallest specialist hospitals
A small hospital with just 3-bed pools	89	48.0%	A small to medium rural or small-town hospital

## Data Availability

All data is publicly available.

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
