# Peer review of "A New Approach for Understanding International Hospital Bed Numbers and Application to Local Area Bed Demand and Capacity Planning"

_ijerph, 2024, doi:10.3390/ijerph21081035_

Round 1
Reviewer 1 Report
Comments and Suggestions for Authors
It's my pleasure to review this manuscript.
Considering the author's compassion for these topics, I would like to express my respect to your work.
Although your good work, I would like to point out something you need to correct and consider in text.
1. Basic but main question.
I know that two countries has different system to use hospital beds. That is, healthcare in England is based on NHS with 'Need' approach' but healthcare in US is based on private, profit with 'Demand' approach. Is it possible to compare and apply the same logic or method?
You did not consider this fact. What's your opinion?
2. If possible, you need to seperate the results and methods or interpretation. In this manucript, some methods, results and interpretations are described in results section.
3. Some figures are wrong.
- line 148; Table S2 -> S1
- In discussion, some list numbers of subtitle are not consecutive.
Author Response
Comments to all reviewers.
Firstly, thank you for your time and valuable comments.
- Note that the Abstract has been modified
- Six additional references have been added
- The Methods section has been expanded
- A number of paragraphs from the Results section have been transferred to the Discussion
- A Further Research section has been added
- A Limitations of the Study section has been added.
It's my pleasure to review this manuscript. Considering the author's compassion for these topics, I would like to express my respect to your work.
Response: Thank you for your encouraging comments.
Although your good work, I would like to point out something you need to correct and consider in text.
1. Basic but main question.
I know that two countries has different system to use hospital beds. That is, healthcare in England is based on NHS with 'Need' approach' but healthcare in US is based on private, profit with 'Demand' approach. Is it possible to compare and apply the same logic or method?
You did not consider this fact. What's your opinion?
Response: To address your question I have added a reference to the work of Brunn et al (2023) who investigated why expressed bed demand is so high in Germany, and why it has been so hard to alter. They outlined several factors specific to Germany. I have added relevant comments in the text.
2. If possible, you need to separate the results and methods or interpretation. In this manuscript, some methods, results and interpretations are described in results section.
Response: This has been attempted.
3. Some figures are wrong.
- line 148; Table S2 -> S1
In discussion, some list numbers of subtitle are not consecutive.
Response: Thank you for identifying these. They have been corrected.
Note that all changes to the text are marked in red.

Reviewer 2 Report
Comments and Suggestions for Authors
The reviewed article, "A New Approach to Understanding the International Number of Hospital Beds and Its Application to Bed Demand and Capacity Planning," proposes a novel model for hospital bed planning based on the Age-Standardized Mortality Rate (ASMR). This is relevant and significant. I think this method is especially useful for identifying discrepancies in bed demand that traditional methods do not capture, as illustrated by the examples of the Northern Territory and Tasmania in Australia.
Strengths:
- The model uses ASMR to predict hospital bed demand, providing a more precise and relevant measure of the population's health needs.
- The model can detect significant differences in bed demand that other methods overlook, allowing for better planning and resource allocation.
Weaknesses:
- The manuscript acknowledges that the model requires further extension to cover specialty-level demand and differentiate between types of patients (elective vs. emergency).
- The high volatility in bed demand introduces uncertainty into the predictions, which could affect the model's accuracy and reliability.
Concerns regarding study design, methodology, or data analysis: The high volatility in bed demand and the dispersion observed around the trend line suggest that there are uncontrolled or unconsidered factors in the model that could influence bed demand, such as changes in health policies, demographic variations, and fluctuations in waiting lists.
Suggestions for improving the clarity or organization of the presentation: To improve the clarity and organization of the manuscript, it is recommended to:
- Improve the structure, starting with the abstract, and adopt a clearer and more defined scientific structure from the beginning.
- Adapt the quality of the writing and ensure compliance with publication standards, as the abstract does not meet the requirements from the start.
- While the results and tables are satisfactory and excellent, the writing lacks justification, or it should be improved, including the background, etc. It is important to clearly justify the study's purpose and enhance it by providing a more significant contribution.
- A better-written methodology explaining how the variables were used in the ASMR model would improve understanding.
- Divide the manuscript into better-defined sections to prevent the results from feeling too lengthy for the reader.
- Provide detailed examples of how the model applies in different contexts to help readers better understand its utility and relevance.
Interesting aspects: The ASMR-based approach has the potential to transform hospital resource planning by providing a more precise measure of bed demand. This could influence future health policies, promoting a more equitable and efficient distribution of hospital resources, and serve as a model for other health systems facing similar challenges.
Ethical implications: The research has significant ethical implications in terms of equity in the distribution of health resources. Addressing and improving this aspect is crucial.
Concerns about generalizability: One concern is whether the model can be generalized to other contexts outside England with different health systems and demographic conditions. The model's reliability could also be affected by the high volatility in bed demand.
Contribution to existing literature or advancement of knowledge: Upon further review, the manuscript makes a significant contribution to the field by introducing an ASMR-based approach to hospital bed planning. This method could address an important gap in the existing literature by providing a more precise and equitable measure of bed demand. Additionally, the approach to evaluating average hospital bed occupancy and its impact on efficiency and safety is noteworthy.
Increasing the sample size and the period analyzed could improve the study's robustness. Moreover, external validation of the model in different contexts and health systems could reinforce the conclusions.
Detailed discussion: Include a more detailed discussion of the study's limitations and how they could be addressed in future research. Review and improve the explanation of external validation to confirm the model's applicability and robustness.
Author Response
Comments to all reviewers.
Firstly, thank you for your time and valuable comments.
- Note that the Abstract has been modified
- Six additional references have been added
- The Methods section has been expanded
- A number of paragraphs from the Results section have been transferred to the Discussion
- A Further Research section has been added
- A Limitations of the Study section has been added.
Changes to the text are in red.
Reviewer 2
The reviewed article, "A New Approach to Understanding the International Number of Hospital Beds and Its Application to Bed Demand and Capacity Planning," proposes a novel model for hospital bed planning based on the Age-Standardized Mortality Rate (ASMR). This is relevant and significant. I think this method is especially useful for identifying discrepancies in bed demand that traditional methods do not capture, as illustrated by the examples of the Northern Territory and Tasmania in Australia.
Strengths:
- The model uses ASMR to predict hospital bed demand, providing a more precise and relevant measure of the population's health needs.
- The model can detect significant differences in bed demand that other methods overlook, allowing for better planning and resource allocation.
Weaknesses:
- The manuscript acknowledges that the model requires further extension to cover specialty-level demand and differentiate between types of patients (elective vs. emergency).
- The high volatility in bed demand introduces uncertainty into the predictions, which could affect the model's accuracy and reliability.
Concerns regarding study design, methodology, or data analysis: The high volatility in bed demand and the dispersion observed around the trend line suggest that there are uncontrolled or unconsidered factors in the model that could influence bed demand, such as changes in health policies, demographic variations, and fluctuations in waiting lists.
Comments added to the text.
Suggestions for improving the clarity or organization of the presentation: To improve the clarity and organization of the manuscript, it is recommended to:
- Improve the structure, starting with the abstract, and adopt a clearer and more defined scientific structure from the beginning.
- Adapt the quality of the writing and ensure compliance with publication standards, as the abstract does not meet the requirements from the start.
- While the results and tables are satisfactory and excellent, the writing lacks justification, or it should be improved, including the background, etc. It is important to clearly justify the study's purpose and enhance it by providing a more significant contribution.
- A better-written methodology explaining how the variables were used in the ASMR model would improve understanding.
- Divide the manuscript into better-defined sections to prevent the results from feeling too lengthy for the reader.
- Provide detailed examples of how the model applies in different contexts to help readers better understand its utility and relevance.
These points have been addressed.
Interesting aspects: The ASMR-based approach has the potential to transform hospital resource planning by providing a more precise measure of bed demand. This could influence future health policies, promoting a more equitable and efficient distribution of hospital resources, and serve as a model for other health systems facing similar challenges.
Ethical implications: The research has significant ethical implications in terms of equity in the distribution of health resources. Addressing and improving this aspect is crucial.
Concerns about generalizability: One concern is whether the model can be generalized to other contexts outside England with different health systems and demographic conditions. The model's reliability could also be affected by the high volatility in bed demand.
Comments added to the text.
Contribution to existing literature or advancement of knowledge: Upon further review, the manuscript makes a significant contribution to the field by introducing an ASMR-based approach to hospital bed planning. This method could address an important gap in the existing literature by providing a more precise and equitable measure of bed demand. Additionally, the approach to evaluating average hospital bed occupancy and its impact on efficiency and safety is noteworthy.
Increasing the sample size and the period analyzed could improve the study's robustness. Moreover, external validation of the model in different contexts and health systems could reinforce the conclusions.
See new section on further studies.
Detailed discussion: Include a more detailed discussion of the study's limitations and how they could be addressed in future research. Review and improve the explanation of external validation to confirm the model's applicability and robustness.
A limitations section has been added
Please indicate if there are aspects that you would like further elaboration.

Reviewer 3 Report
Comments and Suggestions for Authors
Allow me, first of all, to thank you for the opportunity to participate in the review process of this manuscript. The article is interesting and focuses on an important topic from the perspective of more efficient and capable management of available resources.
It is an extensive manuscript, to be sure, but easy to read, capable of retaining the attention of other researchers, but also of managers and political decision-makers, with an interest in the topic. I consider the discussion chapter particularly interesting and well constructed, with particular attention paid to the organization of the different aspects discussed.
If I have to make a recommendation, it concerns the abstract, which is, in my opinion, too long and should be revised to meet the journal's requirements.
Author Response
Comments to all reviewers.
Firstly, thank you for your time and valuable comments.
- Note that the Abstract has been modified
- Six additional references have been added
- The Methods section has been expanded
- A number of paragraphs from the Results section have been transferred to the Discussion
- A Further Research section has been added
- A Limitations of the Study section has been added.
Reviewer 3
Allow me, first of all, to thank you for the opportunity to participate in the review process of this manuscript. The article is interesting and focuses on an important topic from the perspective of more efficient and capable management of available resources.
Thank you for your kind words.
It is an extensive manuscript, to be sure, but easy to read, capable of retaining the attention of other researchers, but also of managers and political decision-makers, with an interest in the topic. I consider the discussion chapter particularly interesting and well constructed, with particular attention paid to the organization of the different aspects discussed.
Thank you.
If I have to make a recommendation, it concerns the abstract, which is, in my opinion, too long and should be revised to meet the journal's requirements.
The Abstract has been revised.
